# Comparison of Infrapatellar and Suprapatellar Intramedullary Nails with New Clinical Score for Fixation of Tibial Shaft Fractures

**DOI:** 10.3390/jfmk10020222

**Published:** 2025-06-09

**Authors:** Giacomo Papotto, Vito Pavone, Gianluca Testa, Rocco Ortuso, Antonio Kory, Enrica Rosalia Cuffaro, Ignazio Prestianni, Emanuele Salvatore Marchese, Saverio Comitini, Alessandro Pietropaolo, Alessio Ferrara, Gianfranco Longo, Marco Ganci

**Affiliations:** 1Department of Orthopaedics and Traumatology, Emergency Hospital Cannizzaro, 95123 Catania, Italy; drrocco@libero.it (R.O.); antonio.kory@gmail.com (A.K.); saveriocomitini@gmail.com (S.C.); giaco84@hotmail.it (G.L.); mrcganci@gmail.com (M.G.); 2Department of General Surgery and Medical Surgical Specialties, Section of Orthopaedics and Traumatology, University Hospital Policlinico G.Rodolico-San Marco, University of Catania, 95123 Catania, Italy; vitopavone@hotmail.it (V.P.); gianpavel@hotmail.com (G.T.); enricacuffaro@outlook.it (E.R.C.); ignazioprestianni93@gmail.com (I.P.); manumarchese1@hotmail.it (E.S.M.); alessandro.pietropaolo@gmail.com (A.P.); alessio.ferrara1993@gmail.com (A.F.)

**Keywords:** tibial fractures, new clinical score, infra–suprapatellar comparison

## Abstract

**Objectives**: Tibial shaft fractures (TSFs) represent the most common diaphyseal fractures in adults. The gold-standard treatment is intramedullary nailing. Recently, the suprapatellar technique has been increasingly adopted due to its ability to reduce complications associated with the infrapatellar approach. Currently, no clinical score for leg fractures comprehensively assesses the entire lower limb. Therefore, we reviewed the main lower-limb scores available in the literature and developed a new clinical evaluation tool for tibial shaft fractures. The aim of our study was to report our experience with both techniques, to compare the outcomes of our prospective study with the international literature, and to propose a new, easy-to-apply, and reproducible clinical score that evaluates the specific functions of the entire lower limb. **Methods**: We conducted a prospective analysis of 920 tibial shaft fractures treated with intramedullary nailing via either a suprapatellar or infrapatellar approach. Patients were divided into two groups: Group A, including 420 patients treated with the infrapatellar approach; Group B, including 500 patients treated with the suprapatellar approach. Follow-up included clinical and radiographic assessments at 1, 3, and 6 months, and annually thereafter. We evaluated differences in patient positioning, operation time, radiation exposure, healing rate, incidence of pseudarthrosis and infection, return to ambulation, residual knee pain and fracture site, persistent lameness, and deformities. For the clinical assessment, we devised a new score—the Catania Hospital Score (CHS)—by integrating the most relevant clinical items from existing lower-limb evaluation tools. The CHS includes anterior knee pain (20 points), lameness (5 points), swelling (10 points), stair-climbing ability (10 points), tibial pain (15 points), the ability to perform daily activities (20 points), and evaluation of deformities (varus/valgus, shortening, rotation, and recurvatum/procurvatum (40 points)), for a total of 120 points. **Results**: Statistically significant differences were observed in Group B regarding a shorter surgical time, a reduced patient positioning time, and decreased radiation exposure. The CHSs were significantly better for Group B at the 3- and 6-month follow-ups. No statistically significant differences were found in infection or pseudarthrosis rates between the two groups. Notably, no cases of chronic knee pain were reported in patients treated with the suprapatellar approach. **Conclusions**: Both surgical approaches are valid and effective. However, our findings indicate that the suprapatellar approach reduces the complications of the infrapatellar technique, improves postoperative outcomes, and does not result in chronic knee pain. The CHS provides a comprehensive, practical, and reproducible tool to assess functional recovery in patients treated with intramedullary tibial nailing.

## 1. Introduction

Tibial diaphysis fractures result from high-energy injuries [1], represent the most common diaphyseal fractures in adults, and account for approximately 2% of all fractures [2]. Their overall incidence is about 16.9–21.5 per 100,000/year [3]. There are various surgical treatment modalities, including open reduction and internal fixation (ORIF), external fixation, and intramedullary nailing (IMN). Intramedullary nail stabilization is the gold standard surgical treatment for diaphyseal fractures of the tibia in adults. Traditionally, an infrapatellar (IP) approach has been used, either through a medial transtendinous incision or a medial or lateral paratendinous incision of the patellar tendon and with the knee flexed more than 90°. Despite the overall excellent results of this approach, well-documented complications include chronic anterior knee pain (47%), malalignment (58–85%), and malunion (30%) [4].

In 1996, Tornetta and Collins introduced a new surgical technique for proximal tibia diaphyseal fractures using a semi-extended knee approach. The approach was developed to reduce the tension force of the patellar tendon on the proximal fragment and reduce the valgus deformities of the proximal third of the tibia typically encountered during the infrapatellar (IP) approach to treat these fractures [5]. This approach is called suprapatellar (SP) access and involves incising the quadriceps tendon. A flexible cannula is then placed in the suprapatellar space and then in the retropatellar space.

Several studies, including recent meta-analyses, show that the suprapatellar (SP) approach may be superior to the IP approach because it allows easier reduction, shorter fluoroscopy times, and reduced rates of malunion and anterior knee pain [6,7,8]. Concerns about the SP approach, however, stem from the introduction of instruments directly into the knee joint [9,10]. Femoral damage is a major preoccupation of the SP approach. Other authors also question an increased risk of knee sepsis, particularly with use in open fractures of the tibial diaphysis [11,12].

A variety of scores for the clinical assessment of the lower limb can be found in the literature, and it was necessary to choose the most suitable score in view of the exact research question and the purpose of this study. In the literature, there is no gold standard for the validation of clinical studies specific to leg fractures that takes into account the whole lower limb examined. For this reason, we unified the main lower limb scores with the highest COSMIN scores [13] and developed our clinical evaluation table for tibial diaphysis fractures.

The aim of our study was to report on our experience of more than 25 years, to describe the differences between the two techniques, to compare the results obtained from our prospective study with the international literature, and to propose a new, easy-to-apply and reproducible clinical assessment system that clinically examines and evaluates the specific functions of the joints around the leg and a full assessment of the lower limb.

## 2. Materials and Methods

We analyzed patients with fractures of the tibial diaphysis treated with an intramedullary nail through a suprapatellar (SP) and infrapatellar (IP) approach. A total of 1255 fractures of the tibial diaphysis were surgically treated. The fractures were classified according to the AO Foundation/Orthopedic Trauma Association’s OTA/AO classification, and we examined diaphyseal fractures classified as 42A-42B-42C. The exclusion criteria were open fractures, articular fractures, fractures with loss of substance, pediatric fractures, patients with knee and ankle joint disease (arthritis–arthrosis), and tibial fractures treated with synthetic means other than an intramedullary nail. Extrapolating the number of cases, we evaluated 920 tibia fractures treated with intramedullary nails, of which 644 were males (70%) and 276 were females (30%), with mean ages of 43.2 and 11.7 years (range: 18–81 years), with a mean body mass index (BMI) of 24.7 ± 6.7, and 54% of the cases (497 patients) involved the right side and 46% (423 patients) the left side. ASA1 included 64 patients (6.9%); ASA2 included 212 patients (23%); ASA3 included 607 patients (65.9%); and ASA4 included 37 patients (4.2%). Of the patients examined, 230 (25%) belonged to subgroup 42A, according to the AO/OTA classification, 400 patients (44%) to subgroup 42B, and 290 cases (31%) to subgroup 42C. The patients were divided into the following two groups: Group A patients treated with the infrapatellar approach (IP) (number: 420 cases); Group B patients treated with the suprapatellar approach (SP) (number: 500 cases) (Table 1). Patients were not randomized.

This was a prospective, comparative clinical study. Patients were consecutively enrolled and assigned to two groups based on the surgical approach used: the infrapatellar (IP) or suprapatellar (SP) approach. The allocation was casual and not influenced by specific clinical or anatomical criteria. The distribution occurred based on operative planning and logistical factors, such as instrument set availability and the operating room schedule, thus limiting potential allocation bias, although no formal randomization protocol was applied. All procedures were performed by experienced trauma surgeons in our department.

For both groups, the pneumatic tourniquet at the thigh root was not used, and spinal-epidural anesthesia was used. For Group A, the patients were placed on the traction table, with transkeletal traction at the calcaneus, support at the popliteal cord, and the knee flexed more than 90°. A median skin incision was made directly at the patellar tendon, and a medial or lateral paramedian incision was made at the patellar tendon. Fracture reduction was often performed by the second operator. For Group B, patients were placed supine on the radiolucent operating table, with their knee semi-extended (with flexion between 20 and 30°). A 2 cm skin incision was made proximal to the upper pole of the patella, a longitudinal incision was made at the quadriceps tendon and joint capsule, and a rubber trocar was inserted inside the knee between the trochlear incisura and the articular surface of the patella to protect the articular salience. Fracture reduction was often performed by the first surgeon (Figure 1 and Figure 2).

The patients were followed up with clinical and radiographic controls at first—in the 3rd–6th months—and every year thereafter from the date of surgery until the clinical and radiological healing of the fracture.

We examined and evaluated the differences in patient positioning, the duration of surgery, the radiation exposure time, the radiographic healing time, the incidence rate of pseudarthrosis and infection, return to walking, and residual knee pain.

The patient positioning time was defined as the time interval (in minutes) between the patient’s entry into the operating room and the completion of the final positioning on the surgical table, just prior to surgical site disinfection.

The radiographic healing time was defined as the number of days elapsed from the date of surgery to the first radiographic evidence of cortical continuity on at least three cortices in standard anteroposterior and lateral views, as independently confirmed by two orthopedic surgeons.

The Catania Hospital Score (CHS) was developed to overcome the limitations of existing lower-limb assessment tools when applied to trauma patients. Numerous clinical scores available in the literature—such as the KOOS, Lysholm, Olerud–Molander, and LEFS scores—are widely used and validated for evaluating joint symptoms, discomfort, and quality of life, primarily in patients with chronic, degenerative, and non-traumatic conditions. These tools are often self-administered and patient-reported, which reduces their utility in the evaluation and follow-up of acute trauma cases, such as tibial shaft fractures treated with intramedullary nailing. To address this gap, we created the CHS by selecting and adapting key questions and scoring systems derived from these validated scales, with the aim of designing a new tool that is simple to apply, easily reproducible, and suitable for a wide range of lower limb pathological conditions. In particular, the CHS integrates trauma-specific parameters—including tibial pain, swelling, and axial or rotational deformities—while maintaining the structured and objective nature of established tools. The CHS evaluates anterior knee pain (max. 20 points), lameness (max. 5 points), swelling (max. 10 points), stair-climbing ability (max. 10 points), tibial pain (max. 15 points), functional autonomy in daily activities (max. 20 points), and the presence and severity of deformities, such as varus/valgus angulation, shortening, rotation, and recurvatum/procurvatum (max. 40 points), reaching a total of 120 points (Table 2). Although formal external validation has not yet been conducted, the CHS was uniformly applied in this study by trained observers. The score was designed to be clinically relevant and practical for orthopedic trauma settings. Interobserver and intraobserver reliability will be assessed in future prospective studies to validate the score’s consistency, applicability, and diagnostic value in different lower-limb pathologies.

### Statistical Analyses

Statistical analyses were performed using the IBM SPSS Statistics software (version 29.0.1.0) for Macintosh. The Shapiro–Wilk test was used to assess whether the samples examined had a normal distribution. Levene’s test was used to check the equality of variance. When the data were normally distributed, and the equality of variance was confirmed, the significance was assessed using Student’s t-test. If the data were not normally distributed, the significance was assessed using Mann–Whitney’s U test. In this study, a *p*-value of <0.05 was considered significant.

To enhance the methodological strength of our study and support the clinical reliability of the newly developed Catania Hospital Score (CHS), we performed a comprehensive statistical validation. The CHS was assessed for internal consistency using Cronbach’s alpha, which demonstrated good homogeneity among items. The interobserver reliability was evaluated through the intraclass correlation coefficient (ICC), confirming reproducibility across the different evaluators. Additionally, the construct validity was explored via factor analysis, supporting the structural coherence of the score components. The discriminant validity was assessed by comparing the CHS values and other clinical–surgical parameters between the two treatment groups (Group A—the infrapatellar approach; Group B—the suprapatellar approach) using the Mann–Whitney test and independent-sample Student’s t-tests. These analyses confirmed the score’s ability to differentiate between distinct surgical techniques. The effect size (ES) was also calculated to estimate the magnitude of the clinical difference in CHS values. To examine the convergent validity, we analyzed the correlation between the CHS and radiographic healing in order to determine whether the CHS reflected functional recovery independently of biological bone union. Lastly, we estimated the minimal clinically important difference (MCID) using the standard method (0.5 × baseline standard deviation), identifying the smallest change in the CHS considered meaningful from a clinical perspective.

## 3. Results

The average time taken to position the patient on the operating table was for Group A, with 7.9 ± 0.96 min per patient, in contrast with Group B, with an average of 5.6 ± 0.75 min per patient (*p* = 0.00001). The mean duration of the intervention was 94.8 ± 9.3 min for Group A and 68.1 ± 8.1 min for Group B (*p* = 0.00001). The mean time of exposure to radiation during surgery was 53.4 ± 7.3 for Group A and 39.6 ± 7.2 for Group B (*p* = 0.00001). The average radiographic healing occurred at approximately 143.8 ± 14.7 days (4.79 months) for Group A, and for Group B, the average radiographic healing occurred at 153.3 ± 11.3 days (5.11 months) (*p* = 0.448067).

Analyzing the data from our CHS rating scale, we obtained no statistically significant results at the clinical check-up 1 month after surgery (*p* = 0.94735). At the clinical check-up in the third month after the date of surgery, we obtained a statistically significant difference between the two groups, with a mean score for Group A of 105.4 ± 7.9 and an average score for Group B of 112.1 ± 7.6 (*p* = 0.00001). At the subsequent clinical follow-up in the sixth month, we also obtained a statistically significant difference between the two test groups, with an average score for Group A of 112.7 ± 6.2 and an average score for Group B of 118.6 ± 1.9 (*p* = 0.00001). At subsequent outpatient follow-ups, no statistically significant data were found between the two groups under investigation, and residual anterior knee pain remained for Group A patients in 130 cases (31%).

We found 9 cases (2.1%) of delayed consolidation 6 months after surgery for Group A and 11 cases (2.2%) for Group B. For Group A, there was only 1 case (0.2%) of pseudoarthrosis at one year from the date of surgery, and for Group B, there were 2 cases (0.4%) of pseudoarthrosis at the clinical check-up at one year. There were 7 cases (1.7%) of superficial soft-tissue infection and 1 case (0.2%) of deep infection for Group A. For Group B, there were 8 cases (1.6%) of superficial skin infection and 2 cases (0.4%) of deep osteomyelitis. We found no statistically significant differences in infection and pseudarthrosis rates between the two groups in our study (Table 3 and Table 4).

To validate the newly developed Catania Hospital Score (CHS), we performed a multi-level statistical analysis. The internal consistency was confirmed with Cronbach’s alpha of 0.81, indicating good homogeneity among the score components. The interobserver reliability was excellent, with an intraclass correlation coefficient (ICC) of 0.86. The construct validity, evaluated through factor analysis, supported the structural coherence of the score. The discriminant validity was demonstrated by statistically significant differences (*p* < 0.00001) in the CHS values, surgical time, and fluoroscopy exposure between Group A (the infrapatellar approach) and Group B (the suprapatellar approach), as assessed by the Mann–Whitney U test and Student’s *t*-tests. The effect size (ES) for the 3-month CHS also confirmed a clinically meaningful difference between the groups. The convergent validity analysis showed no significant correlation between the CHS and radiographic healing (*p* = 0.448), indicating that the CHS primarily reflects functional and symptomatic recovery rather than biological bone healing. The minimal clinically important difference (MCID) was estimated at 6 points, based on 0.5 times the baseline standard deviation (SD = 11.6), and may serve as a reference threshold for interpreting clinical relevance (Table 5).

## 4. Discussion

In our study, we analyzed the clinical–practical differences in the performance of two different surgical techniques, the suprapatellar approach (SP) and the infrapatellar approach (IP). We obtained statistically significant differences in the advantages of Group B, the suprapatellar technique (SP), in terms of the decrease in the patient positioning time, the reduction in the duration of the surgical procedure, and the radiation exposure time. These results can be explained by the use of metal blocks and encumbrances for the infrapatellar (IP) approach, which do not allow a correct intraoperative radiographic visualization of the examined limb; thus, there is a need to change the position of the limb or the shining apparatus, thus increasing the number of radiographs taken to obtain correct images, thus increasing the radiation time and exposure [14]. Sun et al. demonstrated a reduction in the radiation exposure time for the suprapatellar (SP) approach in their work on 162 patients [15], a hypothesis also confirmed by Williamson et al. in their work, where they analyzed 90 tibia nails [16]. In addition, the suprapatellar (SP) approach allows a more accurate entry point of the intramedullary nail measured in anteroposterior and lateral radiographs [17], improves insertion angles [18], and is associated with better alignment rates [19].

Anterior knee pain is a severe complication that affects patients treated with an IP approach. The incidence ranges from 10 to 86% of cases with exacerbation and functional limitation during knee flexion [20]. Although its etiology is unclear, this pain may be associated with injuries to the knee structure and saphenous nerve [21]. Leliveld et al. showed in their study that an IP incision is more likely to cause chronic knee pain due to iatrogenic damage to the infrapatellar nerve after intramedullary nailing [22]. Other studies in the literature show that the IP approach for intramedullary nailing is associated with patellar tendon injuries [23]. The study by Courtney et al. claimed that during the SP approach, the infrapatellar nerve is distant from the incision compared with the IP approach [24]. In the literature, we found several works, including meta-analyses, confirming a lower incidence of anterior knee pain after tibial nailing with the SP approach [8,14]. In our study, we found a statistically significant difference in the CHS clinical evaluation score between the two groups in serial controls at the third and sixth months. In particular, we found no chronic anterior knee pain in Group B (SP) patients, in contrast with patients treated with the IP approach (Group A), where we found an incidence of 31% (130 patients).

The main disadvantage of using the SP approach is the possibility of damage to the articular cartilage. Jakma et al. checked the effect of intramedullary suprapatellar nailing by arthroscopy and observed partial cartilage damage in the femoral trochlea in some cases, but no patients complained of knee joint pain after surgery [25]. Gelbke et al. found that the average contact force on the patellar articular surface was 3.83 mPa through the suprapatellar approach, which was less than 4.5 mPa, the critical pressure for causing chondrocyte injury, so they considered that suprapatellar nailing caused limited cartilage damage [26]. Gaine et al. also showed that the suprapatellar approach was associated with a lower overall incidence of damage to intra-articular structures [27]. Other anatomical studies show similar knee joint damage between the approaches [28].

However, it is important to keep in mind that this new approach is not without complications. One study evaluated 139 exposed tibial diaphysis fractures that were managed with the suprapatellar (SP) approach and showed, in a single case, that septic arthritis of the knee joint can occur after tibial nailing [12]. Another study showed a decrease in the muscle tone of the quadriceps femoris muscle [29].

Another important point in our study was the use of a new clinical evaluation system that allows a comprehensive analysis of all major parameters for the clinical evaluation of the lower limbs. In any study, such as this one, the principal investigator should carefully consider whether only the primary criteria of a clinical assessment score are sufficient for the overall assessment of a patient and quality of life after surgical treatment. Many of the studies in the literature used two or more scores due to the absence of an unambiguous and standardized score to analyze the outcome and prognosis of leg fracture patients. Furthermore, scores designed for use in orthopedic and trauma surgery are used without any validation or validation of clinical scores. The most widely used clinical scores in the literature are the Lysholm Knee Scoring Scale [30], and it is a PROM (patient-reported outcome measure)-type questionnaire. It was developed by Lysholm and Gillquist in Linköping, Sweden, in 1982, from the Larsen Scale and is designed to assess outcomes after knee ligament surgery and, in particular, anterior cruciate ligament reconstruction and symptoms related to instability. Another popular score in the literature is the Oxford Knee Score [31]. It is a PROM-type questionnaire designed for the post-operative self-assessment of patients treated with total knee replacement. The Kujala Anterior Knee Pain Scale (AKPS) [32] is a PROM (patient-reported outcome measure)-type questionnaire, originally published in 1993 by Kujala in Helsinki. It was introduced for patients with patellofemoral disease by assessing subjective symptoms and functional limitations during typical daily activities. The Knee Injury and Osteoarthritis Outcome Score (KOOS) is a self-administered questionnaire, completed by the patient without clinician intervention, which aims to assess reported symptoms at the knee joint in individuals with joint injuries or primary osteoarthritis. None of these instruments are specific for the global post-operative assessment of a treated fractured patient after tibial nailing, which is why many studies in the literature only identified and assessed certain parameters, such as the presence or absence of anterior knee pain [22,33]. Furthermore, the assessment tests in the literature are self-administered questionnaires, i.e., filled in by the patients without intervention by a clinician, with the aim of assessing the symptoms reported by the patients themselves, thus being a highly variable method of assessment. In the case of trauma patients, there is a need for an easy-to-use, reproducible score that takes into account the clinical condition of the patient post-trauma and the post-surgical treatment of a previously healthy patient.

Our evaluation system considers anterior knee pain both at rest and during movement, the presence or absence of lameness during walking, and the presence of swelling in the knee at rest or after activity, and it assesses the ability or inability to climb stairs, analyzes the presence of spontaneous or exacerbated pain following movement at the tibia, and, finally, assesses residual deformities.

Finally, when analyzing the data from our study regarding complications such as infections and pseudoarthrosis, we found no statistically significant data, and, moreover, the data obtained were in line with the international literature [34,35,36].

One of the main contributions of this study is the development and preliminary validation of the Catania Hospital Score (CHS), a novel composite tool specifically designed to quantify early functional recovery following the intramedullary nailing of tibial shaft fractures. Unlike traditional scores, the CHS integrates pain, mobility, weight-bearing capacity, and the need for walking aids into a single clinically oriented metric, enabling a more direct assessment of patient-centered outcomes in the early postoperative phase. To ensure methodological robustness, the CHS underwent a multi-level statistical validation. The internal consistency was confirmed by Cronbach’s alpha of 0.81, indicating strong homogeneity among items. The interobserver reliability was excellent (ICC = 0.86), supporting its reproducibility across different evaluators. The construct validity was explored through factor analysis, confirming the structural coherence of the score components. The discriminant validity was demonstrated by highly significant differences (*p* < 0.00001) between the treatment groups across multiple timepoints, with a strong effect size at three months. Furthermore, the absence of a correlation with radiographic healing (*p* = 0.448) reinforces that the CHS captures functional and symptomatic domains, rather than purely biological recovery. A preliminary estimate of the minimal clinically important difference (MCID) suggested that a variation of approximately 6 points in the CHS may be considered clinically relevant. This enhances the interpretability of future trials and supports the CHS’s application in both clinical practice and research. In light of these findings, the CHS may serve as a promising outcome measure for future comparative studies and clinical audits. Further validation in larger, multicenter cohorts is recommended to confirm its external reliability, responsiveness to change, and cross-cultural adaptability.

### Study Limitations

This study presents several limitations that warrant consideration. First, the monocentric design may limit the generalizability of the findings, as institutional protocols, patient populations, and surgical expertise can influence clinical outcomes. Second, although patients were allocated to treatment groups in a casual and non-selective manner, the absence of formal randomization introduced a potential risk of selection bias. Nonetheless, the baseline clinical and radiographic characteristics were comparable between the groups, which helped mitigate this issue. Third, the blinding of outcome assessors was not implemented due to practical constraints inherent to the clinical setting, potentially introducing observer bias, particularly in the evaluation of subjective outcomes. Regarding the Catania Hospital Score (CHS), although it was newly introduced in this study, several statistical tests were performed to support its validity. The internal consistency (Cronbach’s alpha = 0.81), interobserver reliability (ICC = 0.86), and construct validity via factor analysis were all favorable, and the CHS showed a strong discriminative ability between the treatment groups, with statistically significant differences and large effect sizes. However, external validation across different clinical settings, populations, and languages is still lacking. As such, while the CHS appears to be a promising and robust tool for early functional assessment after tibial nailing, its broader application should be approached with caution until further studies confirm its reproducibility, responsiveness, and generalizability. These methodological limitations largely stem from logistical, ethical, and operational challenges during patient enrollment and follow-up. Future multicenter randomized controlled trials with blinded outcome assessment and standardized protocols—alongside an independent, prospective validation of the CHS—are necessary to confirm and expand upon these preliminary results and to support the score’s integration into routine clinical practice.

## 5. Conclusions

Our study underscores the clinical advantages of the suprapatellar (SP) approach for intramedullary tibial nailing, demonstrating a significantly reduced operative time, superior fluoroscopic control, lower radiation exposure, and a lower incidence of postoperative anterior knee pain compared with the infrapatellar (IP) technique. These findings align with emerging evidence in the literature and support the growing preference for the SP approach in selected clinical contexts, particularly in cases requiring optimal alignment and intraoperative stability. A further strength of this study lies in the development and application of a novel clinician-based scoring tool—the Catania Hospital Score (CHS)—designed to provide an objective and multidimensional assessment of early functional recovery following tibial nailing. The CHS incorporates key clinical parameters, including pain, weight-bearing ability, joint mobility, and local signs, offering a rapid yet structured evaluation suitable for clinical and research settings. Unlike purely radiographic metrics, the CHS emphasizes the patient’s functional status, which is often more relevant in the early postoperative period. To support the validity of this tool, we performed a multi-level statistical validation. The internal consistency was strong (Cronbach’s alpha = 0.81), the interobserver reliability was excellent (ICC = 0.86), and the score demonstrated a high discriminative power between the surgical groups, with large effect sizes and statistically significant differences at three and six months. Although no significant correlation was found between the CHS and radiographic healing, this likely reflects the score’s focus on functional—not biological—recovery. Additionally, we estimated a minimal clinically important difference (MCID) of approximately 6 points, which may serve as a threshold for meaningful clinical change in future studies. Nevertheless, despite these promising results, the CHS has not yet undergone external validation, and its responsiveness, generalizability, and long-term applicability remain to be established. Moreover, while no major complications occurred in the SP group in our cohort, rare but serious events such as intra-articular cartilage injury or joint sepsis have been reported elsewhere. Therefore, careful patient selection and appropriate surgical expertise remain essential to minimize risks. In light of the methodological limitations of our study—including its monocentric, non-randomized design and the absence of blinded outcome assessment—our findings should be interpreted with appropriate caution. Future multicenter randomized controlled trials with standardized follow-up protocols and independent validation of the CHS score are warranted to confirm these preliminary observations and enhance their applicability in daily clinical practice.

## Figures and Tables

**Figure 1 jfmk-10-00222-f001:**
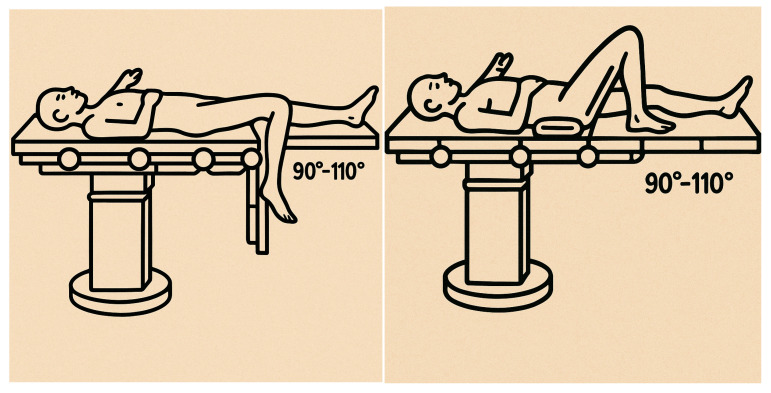
The image shows the positioning of the patient for infrapatellar intramedullary nailing. It is important to note the knee flexion between 90° and 100°, which facilitated access to the tibial canal and fracture alignment. In our study, patients were placed on a traction table using transkeletal traction at the calcaneus. A padded support was placed at the level of the popliteal fossa to maintain stable flexion.

**Figure 2 jfmk-10-00222-f002:**
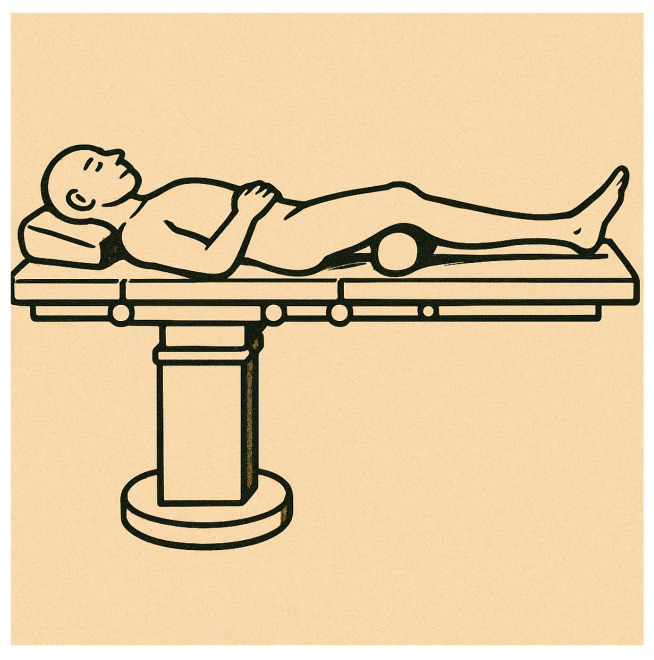
The image shows the patient positioning for suprapatellar intramedullary nailing. Compared with the infrapatellar technique, this positioning was significantly simpler. Patients were placed supine on a radiolucent operating table, with their knee in a semi-extended position (with approximately 10–20° of flexion), allowing easier access to the tibial entry point without the need for deep knee flexion or complex supports.

**Table 1 jfmk-10-00222-t001:** This table shows the design of our study, reporting on the materials and methods.

	Group A	Group B	TOT
**Patients**	420	500	920
**Age**	45.1 ± 10.3	39.8 ± 11.6	43.2 ± 11.7
**Side**	R 215 (51%)	R 282 (63%)	R 497 (54%)
	L 205 (49%)	L 218 (37%)	L 423 (46%)
**Sex**	M 240 (57%)	M 365 (73%)	M 644 (70%)
	F 180 (43%)	F 135 (27%)	F 276 (30%)
**suBMI**	25.1 ± 5.9	22.3 ± 4.2	24.7 ± 6.7
**ASA**	ASA1: 41 (9%)	ASA1: 23 (4.5%)	ASA1: 64 (6.9%)
	ASA2: 107 (26%)	ASA2: 105 (21%)	ASA2: 212 (23%)
	ASA3: 252 (60%)	ASA3: 355 (71%)	ASA3: 607(65.9%)
	ASA4: 20 (5%)	ASA4: 17 (3.5%)	ASA4: 37 (4.2%)
**AO/OTA**	42A: 75 (18%)	42A: 155 (31%)	42A: 230 (25%)
	42B: 210 (50%)	42B: 190 (38%)	42B: 400 (44%)
	42C: 135 (32%)	42C: 155 (31%)	42C: 290 (31%)

**Table 2 jfmk-10-00222-t002:** Catania Hospital Score (CHS).

**Pain anterior knee (max 20 points)**				**Function (max 20 points)**		
A	No pain on walking	10		A	walking and standing unlimited	15
	Mild pain on walking	5			walking distance of 750 m outdoor and standing > 1 h	10
	Severe pain on walking	0			walking up 300 m outdoor and standing < 45 min	6
B	No pain at rest	10			walking inside and brief standing	3
	Mild pain at rest	5			can’t walk	0
	Severe pain at rest	0		B	independent in activities	5
					need for support in activities	0
**Limp (max 5 points)**						
	I have no limp when I walk	5		**Deformity (max 40 points)**		
	I have a slight or periodical limp when I walk	3		A	Varus/Valgus	
	I have a severe and constant limp when I walk	0			none	10
					2°–5°	6
**Swelling (max 10 points)**					6°–10°	3
	I have swelling in my knee	10			>10°	0
	I have swelling in my knee only after vigorous activities	6		B	Anterversion/recurvation	
	I have swelling in my knee after ordinary activities	2			0°–5°	10
	I have swelling constantly in my knee	0			6°–10°	6
					11°–20°	3
**Climbing stairs (max 10 points)**					>20°	0
	I have no problems climbing stairs	10		C	Rotation	
	I have slight problems climbing stairs	6			0°–5°	10
	I can climb stairs only one at a time	2			6°–10°	6
	Climbing stairs is impossible for me	0			11°–20°	3
					>20°	0
**Pain in the tibia (max 15 points)**				D	Shortening	
	I have no pain	15			0–5 mm	10
	I have intermittent pain in the tibia after physical activity	10			6–10 mm	6
	I have marked pain in my tibia during physical activity	8			11–20 mm	3
	I have marked pain in my tibia during/after walking more than 750 mt	4			>20 mm	0
	I have marked pain in my tibia during/after walking less than 750 mt	2				
	I have pain	0		/120 points		

**Table 3 jfmk-10-00222-t003:** This table shows the results of our study.

	Group A	Group B	*p*-Value
**Placement** (min)	7.89 ± 0.94	5.59 ± 0.76	0.00001
**Surgery** (min)	94.84 ± 9.34	68.14 ± 8.06	0.00001
**X-ray exposure** (sec)	53.43 ± 7.3	39.61 ± 7.27	0.00001
**Radiographic healing** (days)	143.8 ± 14.7	153.3 ± 11.3	0.448067
**Nonunion**			
6th month	9 (2.1%)	11 (2.2%)	
1 year	1 (0.2%)	2 (0.4%)	
**Infections**			
Superficial	7 (1.7%)	8 (1.6%)	
Deep	1 (0.2%)	2 (0.4%)	
**CHS**			
1st month	89.9 ± 11.6	95.6 ± 8.3	0.94735
3rd month	105.4 ± 7.9	112.1 ± 7.6	0.00001
6th month	112.7 ± 6.2	118.6 ± 1.9	0.00001
1 year	118.4 ± 1.1	119.1 ± 0.7	0.999

**Table 4 jfmk-10-00222-t004:** A summary of the results and CHS validation metrics.

Parameter	Group A	Group B	*p*-Value
Placement (min)	7.89 ± 0.94	5.59 ± 0.76	**0.00001** *
Surgery (min)	94.84 ± 9.34	68.14 ± 8.06	**0.00001** *
X-ray exposure (sec)	53.43 ± 7.3	39.61 ± 7.27	**0.00001** *
Radiographic healing (days)	143.8 ± 14.7	153.3 ± 11.3	0.44807
Nonunion (6th month)	9 (2.1%)	11 (2.2%)	n.s.
Nonunion (1 year)	1 (0.2%)	2 (0.4%)	n.s.
Superficial infections	7 (1.7%)	8 (1.6%)	n.s.
Deep infections	1 (0.2%)	2 (0.4%)	n.s.
CHS—1st month	89.9 ± 11.6	95.6 ± 8.3	0.94735
CHS—3rd month	105.4 ± 7.9	112.1 ± 7.6	**0.00001** *
CHS—6th month	112.7 ± 6.2	118.6 ± 1.9	**0.00001** *
CHS—1 year	118.4 ± 1.1	119.1 ± 0.7	0.999

* Bold values indicate statistical significance (*p* < 0.01).

**Table 5 jfmk-10-00222-t005:** The preliminary psychometric validation of the CHS.

Validation Domain	Statistical Test/Measure	Result/Value	Interpretation
Internal consistency	Cronbach’s alpha	0.81	Good internal consistency among score items
Interobserver reliability	Intraclass correlation coefficient (ICC)	0.86	Excellent reproducibility across observers
Construct validity	Exploratory factor analysis	Structural coherence confirmed	Supports theoretical structure of the CHS
Discriminant validity	Mann–Whitney U/Student’s t-test	*p* < 0.00001 for CHS, surgical time, and fluoroscopy time	CHS significantly differentiates between surgical techniques
	Effect size (CHS at 3 months)	Large (exact value not reported)	Confirms clinical relevance of score differences
Convergent validity	Correlation: CHS vs. radiographic healing	*p* = 0.448 (non-significant)	CHS reflects functional recovery rather than radiographic consolidation
Minimal clinically important difference (MCID)	0.5 × SD (SD = 11.6)	~6 points	Suggests threshold for meaningful clinical change

## Data Availability

The data are available in the tables in this manuscript.

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
