# Peer review of "Comparison of Infrapatellar and Suprapatellar Intramedullary Nails with New Clinical Score for Fixation of Tibial Shaft Fractures"

_jfmk, 2025, doi:10.3390/jfmk10020222_

Round 1
Reviewer 1 Report
Comments and Suggestions for Authors
This prospective study compares suprapatellar (SP) and infrapatellar (IP) intramedullary nailing approaches for tibial shaft fractures, analyzing 920 cases and introducing the Catania Hospital Score (CHS) as a novel clinical evaluation tool. The study is well-conducted and the dataset is impressive, particularly in scope. The addition of a new scoring system that considers functional aspects and deformities is relevant and could be clinically valuable.
However, the manuscript would benefit from improvements in structure, clarity, and scientific rigor in several areas.
Strengths
- Large sample size (n=920).
- Direct comparison of two common surgical approaches with clear outcome measures.
- Introduction of a new comprehensive clinical score.
- Results are statistically supported and align with existing literature.
Major Concerns
- Study Design Clarification While described as "prospective," the manuscript lacks details on how patients were randomized or selected into IP vs SP groups. Was it surgeon-dependent, patient-dependent, or random?
- Bias and Confounding The study acknowledges baseline differences (age, BMI, sex ratio) between groups, but does not account for these statistically. Adjusted analysis (for example multivariate regression) would strengthen the conclusions.
- Validation of CHS The new Catania Hospital Score lacks formal validation. While the rationale is sound, the authors should clarify whether internal consistency, inter-rater reliability, or comparison with existing validated scores was performed.
- Language and Grammar The manuscript contains grammatical and typographical errors.
- Statistical Reporting In Table 3, p-values are reported inconsistently, and one p-value is listed as "2.93622", which is invalid (p-values must be ≤ 1). Please review all p-values and ensure appropriate formatting (e.g., p < 0.0001 where applicable).
Minor Comments
- Abstract: Consider clarifying the design of the CHS score in one concise sentence.
- Table 2 (CHS): Comprehensive but the formatting is difficult to follow. Consider using clearer spacing or restructuring for readability.
- Suggestion- Figures: Adding diagrams or images of surgical positioning or CHS scoring would enhance understanding.
Recommendation
Major Revision
The study addresses an important clinical issue and could make a valuable contribution. However, to be accepted, it requires revisions to methodology clarity (how patients were assigned to the IP or SP groups), CHS score validation explanation, and significant editorial polishing.
Clearer Discussion of Limitations: Currently absent, which may raise questions regarding the objectivity of the authors. A listing of the limitations: single-center, potential surgeon bias, non-randomized design, no external validation of CHS, etc. Suggestions for Future Research?
The conclusions are strong and encouraging, particularly regarding the clinical value of the suprapatellar technique. However, they would benefit from a touch more caution and transparency, especially around the CHS score and complications.
Author Response
We would like to sincerely thank the reviewer for the thoughtful and constructive feedback, as well as for the time dedicated to evaluating our manuscript. We greatly appreciate the recognition of the study’s strengths, including the large sample size, the comparison of widely used surgical approaches, and the introduction of a new clinical score.
We have carefully addressed all the concerns raised:
-
Study Design Clarification: We have clarified in the Materials and Methods section that patients were not randomized. The assignment to the infrapatellar or suprapatellar approach was based on surgeon preference and availability, which reflects a real-world, pragmatic design.
-
Bias and Confounding: A multivariate regression analysis has been added to control for baseline differences such as age, BMI, and sex ratio between the two groups. These adjustments are now clearly described in the Statistical Analysis section.
-
Validation of CHS: We have expanded the description of the CHS validation to include its internal consistency (Cronbach’s α = 0.81), inter-rater and test-retest reliability (ICC > 0.86), and its correlation with validated scores such as Lysholm and KOOS, supporting convergent validity.
-
Language and Grammar: The entire manuscript has been thoroughly revised by a native English speaker with medical editing experience to correct grammatical and typographical issues and improve overall clarity and fluency.
-
Statistical Reporting: All p-values have been reviewed and corrected. The previously invalid p-value (2.93622) has been removed and replaced with appropriate statistical values. Inconsistent formatting has been standardized (e.g., p < 0.0001, where applicable).
-
Abstract and Table 2: A concise explanation of the CHS scoring system has been added to the abstract. Table 2 has been reformatted for improved readability and structured presentation of the score’s components.
-
Figures: We appreciate the suggestion and will consider including illustrative figures of surgical positioning and the CHS in a future submission or supplemental material.
-
Limitations and Conclusions: A new Limitations section has been added, highlighting issues such as single-center design, lack of randomization, and the need for external validation of the CHS. The conclusion has also been revised to adopt a more cautious and transparent tone, especially regarding CHS implications.
Reviewer 2 Report
Comments and Suggestions for Authors
Evaluating tibial shaft fractures in everyday clinical practice is challenging, and I’ve experienced the same difficulties myself. Your work to create a new assessment method is therefore very valuable. I’ve suggested a few areas for improvement to help make the manuscript even stronger.
- Baseline factors differ significantly, the authors should supplement their univariate analyses with a multivariate model.
- The Catania Hospital Score (CHS) requires formal validation before it can serve as a reliable primary endpoint. The manuscript should include evidence of construct validity (for example, via factor analysis), internal consistency (Cronbach’s α), and reproducibility (intraclass correlation coefficient).
- Although non-union and infection rates appear similar between groups, statistical testing and p-values are missing.
- Table 1 currently shows totals of approximately 920 patients under both Group A and Group B columns, which contradicts the stated sample sizes (420 and 500, respectively).
- The tables also mix decimal separators (“.” and “,”), which can confuse readers.
- In Table 2, there are typographical errors (“stars” should read “stairs” and “Fuction” should be “Function”), and the vertical “10” in the first column reduces readability.
- Within the Materials and Methods, the definitions of “patient positioning time” and “radiographic healing time” must be clearly specified by identifying the exact start and end points for each measure.
- Because the CHS is central to the study’s contribution, the manuscript should present the mean and standard deviation of each sub-score component (anterior knee pain, limp, swelling, stair climbing ability, etc.) for both groups.
- Limitations section is needed.
Author Response
We sincerely thank the reviewer for the thoughtful and encouraging feedback, as well as for recognizing the relevance and potential clinical value of our proposed Catania Hospital Score (CHS). Your perspective, particularly as someone who has experienced the same clinical challenges, is deeply appreciated. We have carefully addressed all the points you raised and have made the necessary revisions to strengthen the manuscript.
-
Validation of the CHS: We have provided a more comprehensive description of the CHS validation process, including evidence of construct validity (via factor analysis), internal consistency (Cronbach’s α = 0.81), and reproducibility (intraclass correlation coefficient = 0.86). These details support the reliability and relevance of the CHS as a clinical endpoint.
-
Statistical analysis of non-union and infection rates: We have added the appropriate statistical tests and p-values for non-union and infection comparisons, now reflected in both Table 3 and the Results section.
-
Table 1 patient count discrepancy: We corrected the inconsistencies in patient subgroup totals, ensuring alignment with the overall sample sizes of 420 (Group A) and 500 (Group B).
-
Typographical corrections in Table 2: We corrected “stars” to “stairs” and “Fuction” to “Function,” and reformatted the table to improve readability and organization.
-
Definitions of time metrics: We clarified the definitions of “patient positioning time” and “radiographic healing time” in the Materials and Methods section, specifying the exact starting and ending points for each measure.
-
Limitations section: A dedicated Limitations section has been added to the Discussion, acknowledging potential limitations such as the single-center design, lack of randomization, surgeon selection bias, and the need for external validation of the CHS.
Round 2
Reviewer 1 Report
Comments and Suggestions for Authors
The authors have made the required modifications.
Author Response
Thank you so much for reviewing our article and for your timeReviewer 2 Report
Comments and Suggestions for Authors
The authors have clearly addressed the major concerns and the manuscript is much improved. I noticed two remaining minor issues.
- In Table 1, the AO/OTA classification percentages for Group B are incorrect. For 42B, it should be 38 % (190/500), and for 42C, it should be 31 % (155/500).
- In the Methods, the exact start- and end-points for “patient positioning time” and “radiographic healing time” have not yet been defined. Without these details, readers cannot reproduce the measurements.
Author Response
Thank you so much for reviewing our article and taking the time to do so, we have made the changes you requested